# Evaluation of Dose Requirements Using Weight-Based versus Non-Weight-Based Dosing of Norepinephrine to Achieve a Goal Mean Arterial Pressure in Patients with Septic Shock

**DOI:** 10.3390/jcm12041344

**Published:** 2023-02-08

**Authors:** Ashley R. Selby, Nida S. Khan, Tara Dadashian, Ronald G. Hall 2nd

**Affiliations:** Department of Pharmacy Practice, Jerry H. Hodge School of Pharmacy, Texas Tech University Health Sciences Center, Dallas, TX 75235, USA

**Keywords:** norepinephrine, vasopressors, dosing, mean arterial pressure, septic shock, critical illness

## Abstract

No consensus exists regarding optimal dosing of norepinephrine in septic shock. We aimed to evaluate if weight-based dosing (WBD) lead to higher norepinephrine doses when achieving goal mean arterial pressure (MAP) than non-weight-based dosing (non-WBD). This was a retrospective cohort study conducted after standardization of norepinephrine dosing within a cardiopulmonary ICU. Patients received non-WBD prior to standardization (November 2018–October 2019) and WBD afterwards (November 2019–October 2020). The primary outcome was the norepinephrine dose needed to attain goal MAP. Secondary outcomes included time to goal MAP, duration of norepinephrine therapy, duration of mechanical ventilation, and treatment-related adverse effects. A total of 189 patients were included (WBD 97; non-WBD 92). There was a significantly lower norepinephrine dose at goal MAP (WBD 0.05, IQR 0.02, 0.07; non-WBD 0.07, IQR 0.05, 0.14; *p* < 0.005) and initial norepinephrine dose (WBD 0.02, IQR 0.01, 0.05; non-WBD 0.06, 0.04, 0.12; *p* < 0.005) in the WBD group. No difference was observed in achievement of goal MAP (WBD 73%; non-WBD 78%; *p* = 0.09) or time until goal MAP (WBD 18, IQR 0, 60; non-WBD 30, IQR 14, 60; *p* = 0.84). WBD may lead to lower norepinephrine doses. Both strategies achieved goal MAP with no significant difference in time to goal.

## 1. Introduction

Norepinephrine is a first-line vasopressor for the treatment of hypotension in critically ill patients with septic shock [1]. Norepinephrine has been associated with a reduction in mortality in randomized, controlled trials when compared to other vasopressor agents [2,3]. Norepinephrine dosing was originally thought to be linear, where a higher dose resulted in a higher mean arterial pressure (MAP) [4]. It has a low volume of distribution and localizes to the sympathetic nervous system tissues, thus it would theoretically not be impacted by body size. Prior studies have found considerable variability in the pharmacokinetics and pharmacodynamics of vasoactive agents, including norepinephrine, between critically ill individuals irrespective of just patient weight [5,6,7,8]. Additionally, obese patients have more adipose tissue, which could cause an increase in adrenergic receptors and, subsequently, an increased requirement of vasopressors for these patients [9]. In theory, larger patients would require larger doses of vasopressors to bind to the excess adrenergic receptors.

There is conflicting evidence regarding weight-based (mcg/kg/min) vs. non-weight-based (mcg/min) norepinephrine dosing. The largest available studies focus only on obese patients, patients at the end of the BMI spectrum (underweight or morbidly obese), or do not compare weight-based vs. non-weight-based dosing approaches [10,11,12].

We conducted a retrospective cohort study to determine if the norepinephrine dose required to achieve a goal MAP of 65 mmHg was significantly different in patients receiving weight-based versus non-weight-based norepinephrine for treatment of presumed septic shock. We hypothesized that patients receiving weight-based dosing would receive significantly higher norepinephrine doses when achieving goal MAP.

## 2. Materials and Methods

We conducted a single-center, retrospective cohort study evaluating the norepinephrine dose requirements (in mcg/kg/min) to achieve a goal MAP of 65 mmHg in critically ill adult patients. We included patients 18 years or older requiring administration of intravenous norepinephrine for presumed septic shock within the cardiopulmonary intensive care unit (ICU) at the North Texas Veterans Affairs Medical Center. Patients were excluded if they received another vasopressor before norepinephrine, received norepinephrine for another indication, or if they received norepinephrine for less than one hour. A norepinephrine dosing policy change was implemented at the institution to help standardize dosing between ICUs. Norepinephrine dosing had previously been prescribed to utilize a non-standardized, non-weight-based approach within the cardiopulmonary ICU, while other areas of the hospital utilized a weight-based approach. After standardization, norepinephrine dosing was standardized to consist of weight-based dosing, using total body weight, beginning at a dose of 0.01 mcg/kg/min, titrated by 0.01 mcg/kg/min every 2 min as needed to achieve a goal MAP of 65–70 mmHg with a maximum of 0.5 mcg/kg/min. Patients in the pre-standardization group (1 November 2018 to 31 October 2019) received non-weight-based norepinephrine dosing. Patients in the post-standardization group (1 November 2019 to 31 October 2020) received weight-based norepinephrine dosing. The electronic health record was updated to make weight-based norepinephrine the only orderable option after the protocol change.

The electronic medical record was used to locate and evaluate patients for inclusion who received norepinephrine during the study timeframe specified. Study patients were identified by review of norepinephrine drug utilization reports and confirmed using an independent chart review comparing patient characteristics and medication charting to the inclusion and exclusion criteria. To obtain an equal number of patients in the weight-based and non-weight-based groups, the data collectors worked in blocks until a convenience sample of approximately 100 patients were included per group.

The following demographics were collected for each patient: age, sex, height, weight on day of norepinephrine initiation, admission diagnosis or primary problem documented, and admission/discharge dates. Laboratory data collected included baseline serum creatine, admission serum creatine, vasopressors, MAP, Glasgow coma score, bilirubin, platelets, mechanical ventilation (including CPAP), PaO_2_ mmHg (ABG), and FiO2%. Severity of illness was measured by calculating the quick Sequential Organ Failure Assessment (qSOFA) score, comorbid conditions were assessed by using the Charleston Comorbidity Index (CCI), and severity of acute illness was assessed using Acute Physiology and Chronic Health Evaluation II (APACHE II). For risk assessment score calculations, the worst value within the first 24 h of ICU admission was used for each patient variable. All scores were calculated using MDCalc© calculators for consistency.

The primary outcome was to evaluate the dose of norepinephrine in mcg/kg/min required to attain a goal MAP of 65 mmHg in critically ill patients receiving weight-based versus non-weight-based doses for septic shock. Non-weight-based doses were converted to weight-based doses using patients’ documented weights to allow for comparisons between groups. Secondary outcomes included the time until achievement of goal MAP, duration of norepinephrine therapy, duration of mechanical ventilation, development of acute kidney injury (AKI) and need for continuous renal replacement therapy (CRRT) in the ICU, ICU length of stay, hospital length of stay, and in-hospital mortality. Patients with an increase in serum creatinine of at least 0.3 mg/dL from baseline or at least a 50% increase in serum creatinine were deemed as having an AKI. Patients with known end-stage renal disease require dialysis (hemodialysis or peritoneal) were not counted as requiring CRRT because they were presumably not receiving CRRT for a new kidney injury. The occurrence of adverse effects related to norepinephrine—including arrhythmias (e.g., atrial fibrillation), myocardial infarction, new infectious episodes—was also evaluated.

Baseline, demographic, and outcome variables were compared between the weight-based and non-weight-based norepinephrine dosing groups. Categorical variables were compared using Fisher’s exact or χ^2^ tests. Continuous data were analyzed using the Mann–Whitney test. All analyses were performed by investigators using STATA (Stata Corp, College Station, TX, USA). This study was approved by the Institutional Review Boards of the Veterans Affairs North Texas Health Care System and the Texas Tech University Health Sciences Center.

## 3. Results

A total of 246 patients screened and 189 were included in the cohort (Figure 1).

Baseline characteristics were similar between the two groups (Table 1). The cohort was comprised of patients who were primarily Caucasian males over 60 years of age. The interquartile range for patient weight was approximately 70 to 100 kg for both groups. Thirty-one percent of the cohort was obese (BMI ≥ 30). There was also no difference in APACHE II scores or the use of additional vasopressors between the groups. Weight-based patients were more likely to have higher qSOFA scores. The non-weight-based group had statistically significantly higher baseline BUN and SCr values prior to admission, but neither of these differences appears to be clinically relevant.

Approximately 26% of patients in each group were never below the goal MAP. The goal MAP of 65 mmHg was achieved for a similar portion of patients who were initially below goal MAP whether they received weight-based or non-weight-based dosing (100% vs. 96%, *p* = 0.11). Three patients were never below the MAP goal in the non-weight-based group while in the ICU. The norepinephrine dosing information for patients who achieved their goal MAP is shown in Table 2. The median norepinephrine dose (mcg/kg/min) required to reach the goal MAP was significantly lower in the weight-based group vs. the non-weight-based group (0.05 vs. 0.07 mcg/kg/min, *p* < 0.001). When compared in mcg/min, the median doses were also lower for patients receiving a weight-based infusion (4.18 vs. 5.30 mcg/min, *p* < 0.001). The median time to goal MAP was 30 min for both groups. The median total days of norepinephrine therapy were longer in the weight-based group (3 vs. 2, *p* = 0.002). This difference was also observed when evaluating the entire cohort.

Table 3 shows the dosing information and clinical outcomes for all patients. Significantly lower starting doses of norepinephrine were used in the weight-based group (0.02 vs. 0.06, *p* < 0.001). The duration of mechanical ventilation, development of AKI, ICU length of stay, hospital length of stay, or in-hospital mortality did not differ significantly between groups. However, more patients in the weight-based group required CRRT during their admission.

## 4. Discussion

This study was designed to determine whether our facility’s practice change to standardized weight-based dosing of norepinephrine in the cardiopulmonary ICU resulted in higher doses of drug administration compared to non-weight-based dosing in order to achieve the desired effect. We found that patients receiving weight-based norepinephrine received lower doses in order to achieve the MAP goal. This was consistent regardless of if the dosing units were based on weight or not. Weight-based dosing also resulted in lower initial doses that did not prolong the time to MAP goal.

In 2016, the United States Food and Drug Administration provided funding to the American Society of Health-System Pharmacists in order to develop and implement standardization of medication concentrations to reduce errors and improve transitions of care. Standardize 4 Safety is the first national interprofessional patient safety initiative to standardize medication concentrations. The first set of concentration recommendations including infusion concentrations of various medications as well as dosing units to be used for the administration of each infusion were released in 2020. Standardize 4 Safety specifically recommends weight-based dosing of norepinephrine.

However, there is no current consensus about which norepinephrine dosing strategy should be the standard of care. Landmark trials documenting the need to utilize norepinephrine as a first-line vasoactive agent in septic shock utilized different dosing strategies [2,13]. Specifically, one was weight-based and one was not. A 2019 survey of 223 critical care pharmacists (~13% response rate) found that 38% used weight-based dosing, 60% used non-weight-based dosing, and 2% said either strategy could be ordered at their institution [14]. Therefore, equipoise remains in determining the optimal dosing norepinephrine strategy for septic shock.

Arabi et al. noted in a retrospective evaluation of 2882 patients from 28 medical centers over 12 years that patients with a BMI of 40 kg/m^2^ or more had a higher mean MAP (62.6 vs. 58.0 mmHg) and lower mean heart rate (112.2 vs. 116.7 beats per minute) on admission than patients with a BMI of 25–29.99 kg/m^2^ [15]. The mean norepinephrine requirement at six hours ranged from 0.30–0.39 for patients with a BMI < 40 kg/m^2^. The omnibus *p*-value of 0.05 for this comparison was likely driven by the mean dose of 0.18 mcg/kg/min for patients with a BMI of 40 kg/m^2^ or more.

A later retrospective cohort study evaluating 100 non-obese (median BMI 24, IQR 21, 26) and 100 obese (median BMI 36, IQR 32, 42) patients with septic shock received lower weight-based doses of norepinephrine compared to non-obese patients (0.09 mcg/kg/min vs. 0.12 mcg/kg/min, *p* = 0.05) [10]. Patients in this study required similar total norepinephrine doses (9 mcg/min vs. 8 mcg/min, *p* = 0.59). The doses utilized in our cohort were approximately half. In addition, the norepinephrine dose and attainment of a MAP of 65 mmHg was assessed at 60 min. We have no way to evaluate the doses or MAP attainment rates at 30 min, which was the median MAP attainment time for our cohort.

Vadiei et al. assessed time to goal MAP in a retrospective cohort of 287 obese patients with septic shock receiving weight-based or non-weight-based norepinephrine dosing [11]. There was no difference observed in the median time to goal MAP regardless of whether weight-based or non-weight-based norepinephrine dosing was used (58 vs. 60 min, *p* = 0.28). The authors noted that weight-based dosing might lead to a longer median duration of norepinephrine administration (33 vs. 27 h, *p* = 0.03) and higher median cumulative norepinephrine doses (12.6 vs. 10.5 mg, *p* = 0.04). The findings of higher cumulative doses and higher doses at 24 h are not surprising because patients in the weight-based group received norepinephrine for a longer duration. The weight-based group did have lower MAP values at 24 h, but the clinical impact of these differences is unclear. Regardless, it appears that clinicians did not feel as comfortable with the weight-based dosing and kept patients on norepinephrine longer. Our study included obese and non-obese patients, but our cohort was otherwise similar to this study cohort. APACHE II scores for our patients were lower than what may be expected for critically ill patients with septic shock, but are similar to the Vadiei cohort.

A large historical cohort study of 2017 patients from Mayo Clinic ICUs from 2010–2015 found that morbidly obese (25%) patients had the lowest in-hospital mortality rates when compared with normal weight (28%) and underweight (41%) patients (*p* < 0.001) [12]. However, this difference appears to be primarily driven by the larger mortality rate in the underweight group. The morbidly obese group also had the highest median cumulative exposure to norepinephrine (underweight = 5.4 mg, normal weight = 5.2, morbidly obese = 9.1 mg, *p* < 0.001). Though the authors note the high cumulative exposure, they do not specify the infusion durations for each group. Total norepinephrine exposure was an independent predictor of mortality and was also associated with longer hospital lengths of stay and greater incidence of AKI and cardiac arrhythmias. While peak dose in mcg/min was also highest in the morbidly obese group, mcg/kg/min was identical between the normal weight and obese groups. However, morbidly obese patients had reduced in-hospital mortality compared to other cohorts. Interestingly, class I and II patients with obesity (BMI 30–39.99 kg/m^2^) were excluded from this cohort. This cohort may have been more acutely ill than our patients.

In sum, these previous results paint a confusing picture of the optimal approach to dosing norepinephrine in patients with septic shock. It feels like the various groups have evaluated different portions of the patient population, yet none have evaluated all of the potential incoming patients. Our study is the first to try to evaluate both non-obese and obese patients receiving norepinephrine for septic shock to determine the impact of this practice change on all of our patients. Our finding of lower dose requirements for MAP attainment for both non-obese and obese patients using weight-based dosing suggests that this dosing strategy may be safe and effective for all patients.

We could not assess multiple important details within this study because of the retrospective design and incomplete documentation within the electronic health record. This means that we were unable to determine indications for patients initiated on CRRT during their admission. Possible reasons for this could include patient/family wishes, baseline need for dialysis, or differences in provider practices. The clinical significance of this result is unclear given the lack of impact on other outcomes, including ICU and hospital length of stay as well as hospital mortality. Unfortunately, this is a very common issue for retrospective studies of ICU patients. Many studies, including ours, were unable to fully assess cumulative norepinephrine doses, total time until vasoactive agents were discontinued, as well as when other agents were added and stopped [10,11,12]. Vadiei et al. had the most robust data collection but still noted the collection of safety-related outcomes as challenging and a limitation [11]. This was a common issue for the other studies in this area as well, including ours. Wong et al. specifically sought to evaluate the impact of norepinephrine dosing on tachycardia and still were unable to assess for many safety issues [16]. Kotecha et al. were able to report fluid balance data, but did not report data regarding fluid administration [12]. We were also unable to capture reliable data regarding fluid administration data because the results we found were much lower than what have been observed in practice. Several studies, including ours, did not include data regarding antibiotic use or source of infection [10,12]. Some electronic medical records now provide a field code for the suspected source of infection as part of their process to order antimicrobial agents, thus this limitation may lessen over time if these data are verified to be of sufficient detail and accuracy.

This issue does not only pertain to past reports regarding norepinephrine dosing regimens. The eICU database reported that 53% of hospitals either did not report or had low (0–20%) reporting for the *infusionDrug* category, which would include a titratable drug like norepinephrine [17]. Similarly, many fluids may be reported as non-specific names such as “Crystalloids (mL)|Continuous infusion meds” for the *intakeOutput* table. We are cautiously optimistic that these reporting issues will improve over time given the increasing attention being paid to the importance of real-world evidence.

The generalizability of our findings to others has some additional limitations beyond the general issues that have impacted the field at large, which we have outlined above. This study being conducted retrospectively at a single center makes it possible that other practice patterns at our facility may have biased our findings. Specifically, our cohort being primarily white, male, elderly veterans may not accurately represent the patient populations at other institutions. Our cohort was smaller than others that evaluated norepinephrine dosing in obese patients. However, our study is the first to specifically evaluate the impact of a norepinephrine dosing change in a pre-/post-study design fashion that has included all patients, regardless of their body size.

## 5. Conclusions

A standardized weight-based dosing strategy yielded lower norepinephrine doses than non-weight-based dosing in our cohort of critically ill patients with presumed septic shock. This conclusion is limited by its retrospective design and single-center nature. Because weight-based dosing provides lower norepinephrine exposures that achieved the goal MAP at a similar timepoint without increasing LOS or mortality, this approach appears to be safe and effective, but confirmatory studies are needed.

## Figures and Tables

**Figure 1 jcm-12-01344-f001:**
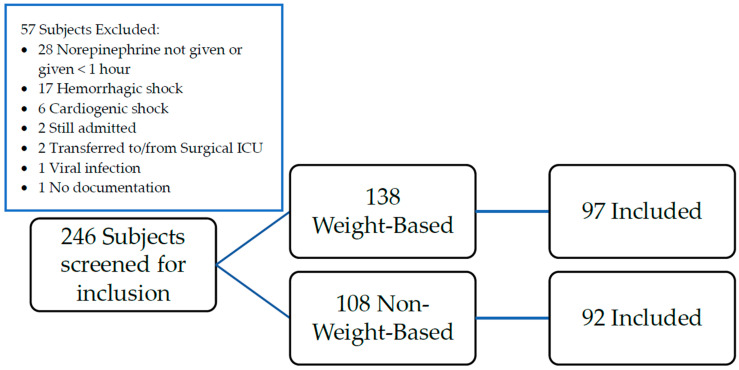
Patient Enrollment. A total of 97 patients received weight-based norepinephrine while 92 patients received non-weight-based norepinephrine; 57 patients were excluded. The most common reasons for exclusion were receipt of norepinephrine for less than one hour (n = 28) or having hemorrhagic shock (n = 17).

**Table 1 jcm-12-01344-t001:** Demographic and Baseline Characteristics.

Characteristic	Weight-Based (n = 97)	Non-Weight-Based (n = 92)	*p* Value
Age (years), median [IQR]	71 [63–75]	71 [63.5–75]	0.99
Male, n (%)	91 (93.8%)	90 (97.8%)	0.17
Weight (kg), median [IQR]	83.4 [69.7–99.1]	81.6 [69.3–100.4]	0.95
Obese (Body Mass Index ≥ 30 kg/m^2^), n (%)	29 (29.9)	29 (31.9)	0.77
Race, n (%)			0.23
Caucasian	55 (56.7)	59 (64.1)	
African American	38 (39.2)	31 (33.7)	
Latino	4 (4.1)	2 (2.2)	
Admission source, n (%)			0.33
Emergency Department	79 (81.4)	77 (83.7)	
Outside Hospital	10 (10.3)	5 (5.4)	
Outpatient/Clinic	6 (6.2)	10 (10.9)	
Other	2 (2.1)	0 (0)	
qSOFA, n (%)			0.03
0	10 (10.3)	23 (25.0)	
1	50 (51.6)	42 (45.7)	
2	26 (26.8)	23 (25.0)	
3	11 (11.3)	4 (4.3)	
qSOFA, median [IQR]	1 [1–2]	1 [0.5–2]	0.02
APACHE II score, median [IQR]	18 [13–23]	16.5 [12.5–24]	0.79
Charlson Comorbidity Index, median [IQR]	7 [5–9]	6 [5–8]	0.23
Charlson Comorbidity Index % 10-year survival, median [IQR]	0 [0–21]	2 [0–21]	0.54
Initial lactate (mmol/L), median [IQR]	2.3 [1.3–3.7]	2.3 [1.2–4.8]	0.54
Intravenous fluid resuscitation documented, n (%)	34 (35)	39 (42.4)	0.19
Intravenous fluid given, n (%)			
0.9% Sodium chloride	19 (19.6)	20 (21.7)	0.72
Lactated ringer’s	16 (16.5)	19 (20.7)	0.58
Albumin	0 (0)	2 (2.1)	0.17
First MAP documented in ICU (mmHg), median [IQR]	79 [65–93]	78 [67–95.5]	0.57
Last MAP when norepinephrine started (mmHg), median [IQR]	58 [54–64]	59 [53–64]	0.86
SCr (mg/dL) at baseline (prior to admission), median [IQR]	1.02 [0.82–1.24]	1.13 [0.91–1.50]	0.05
BUN (mg/dL) at baseline (prior to admission), median [IQR]	15 [11–21]	18 [13–29]	0.01
SCr (mg/dL) on admission to ICU, median [IQR]	2 [1.23–3.38]	1.68 [1.19–2.9]	0.22
BUN (mg/dL) on admission to ICU, median [IQR]	32 [19–54]	29.5 [19–47.5]	0.43
Additional agents, n (%)	38 (39.2)	44 (47.8)	0.23
Vasopressin	29 (29.9)	37 (40.7)	
Epinephrine	12 (12.4)	13 (14.3)	
Dopamine	2 (2.1)	6 (6.5)	
Phenylephrine	2 (2.1)	2 (2.2)	
Dobutamine	5 (5.2)	4 (4.3)	
Milrinone	0 (0)	6 (6.5)	

Abbreviations: APACHE = Acute Physiology and Chronic Health Evaluation; BUN = blood urea nitrogen; dL = deciliters; IQR = interquartile range; kg = kilograms; L = liters; m = meters; MAP = Mean Arterial Pressure; mmHg = millimeters of mercury; mg = milligrams; mmol = millimoles; n = number of patients; SCr = serum creatinine; qSOFA = quick sequential organ failure assessment; % = percentage of patients.

**Table 2 jcm-12-01344-t002:** Results of Patients who Achieved the Mean Arterial Pressure Goal.

	Weight-Based (n = 71)	Non-Weight-Based (n = 65)	*p*-Value
Norepinephrine dose at goal MAP (mcg/kg/min), median [IQR]	0.05 [0.02–0.07]	0.07 [0.05–0.13]	<0.001
Norepinephrine dose at goal MAP (mcg/min), median [IQR]	4.18 [1.52–5.91]	5.30 [4.04–11.56]	<0.001
Time to goal MAP (min), median [IQR]	30 [15–60]	30 [15–60]	0.75
Duration of norepinephrine (days), median [IQR]	3 [2–4]	2 [1–3]	0.002

Abbreviations: IQR = interquartile range; kg = kilograms; MAP = Mean Arterial Pressure; mcg = micrograms; min = minute(s); n = number of patients; % = percentage of patients.

**Table 3 jcm-12-01344-t003:** Results.

	Weight-Based (n = 97)	Non-Weight-Based (n = 92)	*p*-Value
Initial norepinephrine dose (mcg/kg/min), median [IQR]	0.02 [0.01–0.05]	0.062 [0.04–0.12]	<0.001
Duration of norepinephrine (days), median [IQR]	2 [2–4]	2 [1–3]	0.02
Duration of mechanical ventilation (days), median [IQR]	2 [0–7]	2 [0–5.5]	0.57
Incidence of acute kidney injury, n (%)	71 (50.7)	69 (49.3)	0.86
Required CRRT during admission, n (%)	26 (26.8)	11 (12)	0.01
ICU Length of stay (days), median [IQR]	7 [4–13]	9 [5–16.5]	0.21
Hospital length of stay (days), median [IQR]	15 [8–29]	15.5 [8.5–26]	0.94
In-hospital mortality, n (%)	44 (45.4)	49 (53.3)	0.28

Abbreviations: CRRT = continuous renal replacement therapy; ICU = intensive care unit; IQR = interquartile range; kg = kilograms; MAP = Mean Arterial Pressure; mcg = micrograms; min = minute(s); n = number of patients; % = per-centage of patients.

## Data Availability

The data that support the findings of this study are available from the corresponding author, A.R.S., upon reasonable request.

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
