# Peer review of "Evaluation of Dose Requirements Using Weight-Based versus Non-Weight-Based Dosing of Norepinephrine to Achieve a Goal Mean Arterial Pressure in Patients with Septic Shock"

_jcm, 2023, doi:10.3390/jcm12041344_

Round 1
Reviewer 1 Report
Selby and co-authors present data on a standard vs weight-based norepinephrine dose in patients with septic shock. The clinical question is relevant and the data basis propitious, so that exploiting it is of benefit to the community. I do have a few comments and questions about the analysis and interpretation.
1. Page 2, lines 68-70. The two data sets do not have a gap between them in time though it is hard to believe that a clinic completely changed its clinical practice from Oct 31 to Nov 1. Either consider leaving a gap or please discuss to what extent the overlap period may affect results (are the data different in November 2018 compared to October 2019 for example?).
2. Fig. 1: 41/138 patients were excluded in the weight-based group and 16/108 in the non-weight-based. Please explain this large difference, which awakens concerns of bias.
3. Page 3, line 122 and Table 1: The sentence states that there was no difference in qSOFA score, though the difference is formally significant. The medians provided in Table 1 are equal. This demonstrates that the presentation of these qSOFA data in Table 1 is inadequate. Please list the number of patients with scores 0, 1, 2, 3, >3 or similar. The formal test can be performed on this contingency table as well as the assessment of whether or not a formal statistical difference is clinically meaningful.
4. Page 3, line 122: Briefly mention SCr and BUN at baseline, which are also formally significant and assess again the clinical relevance. Start a new paragraph before beginning presentation of outcome variables (“The goal MAP…”).
5. Page 4 line 132 and Table 2: In the text you note that “need for CCRT during admission” “did not differ significantly between groups” but in Table 2 there were 26.8% vs 12% of patients who “Required CRRT during admission” (p=0.01). I verified that the p-value is correct. Please modify and discuss.
6. Table 1: “BUN” is not listed as an acronym (blood urea nitrogen surely) and the unit (mg/dl) should be provided.
7. Table 1: There is a layout problem in the block “Admission source”. The columns do not line up correctly.
8. Table 1: Admission source for outpatient/clinic in the non-weight based group should read 10 (10.9) instead of 10 (62.5).
9. Table 2: “Time to goal MAP” is quite different between the arms (median 18 min vs 30 min). First off, note that the analysis is inherently problematic since it is unclear how to deal with patients that do not achieve the goal (were they simply left out of this analysis?). Since it was 71 vs 72 patients, this will unlikely cause bias in this particular data set. Add the Hodge-Lehmann estimate for the difference in pseudo-median along with a 95% CI interval so and please recheck the result. It is possible that there is a meaningful estimated difference and that the analysis is underpowered to detect it. It would be a pity to overlook this. Also add a t-test (parametric analysis) to compare means – this may be most appropriate on a log scale, where 1 may have to be added to the times to avoid technical issues. The central limit theorem shows that such an analysis is likely appropriate in this context despite the non-normal distribution of the data.
10. Table 2: The p-value for “achievement of goal MAP” does not math the data (should be 0.42).
11. Table 2: “Duration of norepinephrine” differs, but it is not entirely clear in which direction and if this is in part merely due to change in clinical rules/policy. Please add more details.
12. Page 6, line 172: Citing paper [10] the claim is that “patients with septic shock required lower weight-based doses”. Do you mean “required” or “received”? Please note the MAP goal if provided in that paper.
13. Page 6, line 185: In the discussion of median cumulative norepinephrine doses, please also discuss why doses at 18-24 h were so different between the groups in Vadiei et al [11].
Reviewer 2 Report
This is an interesting preliminary results about weight-based vs non-weight-based dosing of norepinephrine. The limitations of the study are clearly stated and exploratory statistics are clearly presented.
The authors cited a paper about obese having higher mean MAP, did the authors observed the same in their cohort?
Line 177 is confusing. What was the the MAP attainment time for your cohort? Were you able to evaluate the doses or MAP attainment within 3o minutes or not? Please clarify.
